# Achievement of Low-Density Lipoprotein Cholesterol Targets in Cardiac Rehabilitation: Impact of the 2019 ESC/EAS Dyslipidaemia Guidelines

**DOI:** 10.3390/jcm11237057

**Published:** 2022-11-29

**Authors:** Matthias Haegele, Aleksandar Djurdjevic, Fabian Jordan, Yu-Ching Liu, Leonie Mildner, Simon Frey, Ivo Strebel, Olivier Clerc, Thilo Burkard, Otmar Pfister

**Affiliations:** Department of Cardiology, Cardiovascular Research Institute Basel (CRIB), University Hospital Basel, University of Basel, 4031 Basel, Switzerland

**Keywords:** cardiac rehabilitation, secondary prevention of cardiovascular disease, lipid lowering therapy

## Abstract

In 2019 the European Society of Cardiology (ESC) lowered the target values for low-density lipoprotein cholesterol (LDL-C) from <1.8 mmol/L to <1.4 mmol/L for secondary prevention of cardiovascular disease (CVD). The aim of this study was to determine the clinical impact of the 2019 ESC/EAS dyslipidaemia guidelines on lipid-lowering therapies and achievement rates of LDL-C targets in a contemporary cohort of CAD patients participating in an ambulatory cardiac rehabilitation (CR) program.We conducted a retrospective analysis of prospectively collected data from the Swiss Secondary Prevention Registry (SwissPR) in patients with Coronary Artery Disease (CAD), who completed the ambulatory cardiovascular rehabilitation program (CR) of the University Hospital Basel, Switzerland from January 2017 to April 2021. To evaluate the impact of the guideline publication, the cohort was split into a pre-Guideline 2019 group (A) and a post-Guideline 2019 group (B). In total 1320 patients were screened leaving 875 patients for analysis. At discharge, more patients in group B were on maximal statin doses (20% vs. 9%, *p* < 0.0001) and on combination therapy with ezetimibe (51% vs. 17%, *p* < 0.0001) than in group A, which resulted in 53% of patients reaching the LDL-C target of <1.4 mmol/L in group B. Regression analysis revealed that dyslipidaemia and positive smoking history represent independent predictors for intensified lipid-lowering medication, whereas absolving CR after publication of the 2019 guidelines was the only significant predictor for reduced LDL-C at CR discharge. We found a significant difference in prescription rates of lipid-lowering medication, especially combination therapies and statin doses, after publication of the 2019 ESC/EAS dyslipidaemia guidelines resulting in an achievement rate of >50% of the LDL-C target <1.4 mmol/L in CAD patients participating in ambulatory CR.

## 1. Introduction

Despite significant progress in secondary prevention of coronary artery disease (CAD) in previous decades, recurrent cardiovascular events remain a major burden for health care systems and the society as a whole [1,2]. Low-density lipoprotein cholesterol (LDL-C) is one of the major modifiable risk factors for ischemic cardiovascular events and a pivotal therapeutic target in secondary cardiovascular prevention. Reducing LDL-C by only 1 mmol/L during five years results in a reduction of major cardiovascular events by >20% [3]. Recent advances in lipid-lowering therapies including the use of high-potent statins and their combination with non-statin medications such as ezetimibe and PCSK9 inhibitors resulted in LDL-C reduction of 50–80% and improved clinical outcomes [4,5]. Given the linear association of LDL-C levels with cardiovascular events the European Society of Cardiology (ESC) and the European Atherosclerosis Society (EAS) lowered the LDL-C target values from <1.8 mmol/L (<70 mg/dL) to <1.4 mmol/L (<55 mg/dL) for secondary prevention of cardiovascular disease (CVD) in their 2019 dyslipidaemia guidelines [6]. Knowing that a large proportion of patients already fail to reach the 2016 guideline LDL-C targets [7,8,9,10], the latest guideline-update represents an ambitious challenge for LDL-management in daily clinical practice.

The aim of this study was to determine the clinical impact of the 2019 ESC/EAS dyslipidaemia guidelines on lipid-lowering therapies and achievement rates of LDL-C targets in a contemporary cohort of CAD patients participating in an ambulatory cardiac rehabilitation (CR) program.

## 2. Materials and Methods

### 2.1. Study Design and Population

This is an observational retrospective cohort study performed using prospectively collected data from the Swiss Secondary Prevention Registry (SwissPR). SwissPR includes all patients undergoing an ambulatory cardiac rehabilitation program at the University Hospital of Basel, Switzerland, a tertiary referral center covering the northwestern part of Switzerland. Patient referral for this program is based on referrals by the University Hospital Basel, local and regional hospitals and referrals by treating general practitioners and cardiologists. The CR consisted of a 12-week multidisciplinary intervention comprising physical training (strength, endurance, coordination) as well as patient education for risk factor management with smoking cessation support as well as nutritional counselling. We included consecutive patients who completed the CR program between January 2017 and April 2021 after a confirmed diagnosis of CAD. Exclusion criteria were not having CAD, not having completed the first three weeks of the 12-week CR program, missing clinical data for study endpoints, or not having signed the informed consent. The study was approved by the local ethics committee (Ethikkommission Nordwest- und Zentralschweiz, approval number EC 2019-02057) and conducted in accordance with the Declaration of Helsinki. All authors were involved in the study design and had access to the data to vouch for the accuracy and completeness of the data analyses.

### 2.2. Lipid Measurements and Lipid-Lowering Therapies

Plasma levels of total cholesterol, low-density (LDL-C) and high-density lipoprotein (HDL-C) cholesterol and triglycerides were measured at CR entry and at discharge. LDL-C was calculated using the Friedewald formula [11]. Lipid-lowering medication was recorded at entry and at discharge of CR. Changes in lipid-lowering therapies during the CR program were made at the discretion of the treating physician. Atorvastatin and Rosuvastatin were classified as high-potent statins. Dosage of high-potent statins is stated as absolute number and relative percentage of maximum daily dosage (MDD); both 40 mg Rosuvastatin and 80 mg Atorvastatin were considered as 100% MDD. For the analysis, all single- and double-agent drugs containing ezetemibe, that are commonly available in Switzerland were considered.

### 2.3. Study Endpoints

The following endpoints were analysed and compared between patients that finished CR before and after the publication of the 2019 ESC/EAS dyslipidaemia guidelines: (1) Intensity of lipid lowering therapy, measured as % of potent statin use and % of maximal tolerated dose prescribed and % of combination therapy at CR entry and at discharge; (2) achievement rates of the ESC 2019 LDL-C guideline target (<1.4 mmol/L; <55 mg/dL) and (3) achievement rates of the ESC 2016 LDL-C guideline target (LDL < 1.8 mmol/L; <70 mg/dL).

### 2.4. Statistical Analysis

Non-parametric data is described as medians and interquartile ranges (IQR). Distribution of continuous variables was compared with Kruskal–Wallace test, distribution of ordinal variables was compared with the Wilcoxon Rank Sum Test, categorical variables were tested with Chi-Square Test and Fisher`s Exact Test as applicable. An alpha level of 0.05 was considered significant. Univariate Multivariable Regression analysis was used to identify predictor variables for three different outcomes, thus three models where computed. Eligible variables comprised of patients’ characteristics, rehabilitation characteristics and clinical relevant characteristics. Thereby difference in both groups and guideline validity period was considered. The first was to search for predictors for an elevated statin dose, in this logistic regression model a Dummy Variable was calculated defining 1 as a higher statin dose prescribed at discharge compared to entry. The second also was a logistic regression model trying to predict whether ezetimibe therapy was started during CR, again a Dummy Variable was calculated defining 1 for all patients who had ezetimibe prescribed at discharge but not at entry. The third was a quantile regression model trying to predict LDL-C values at discharge. To assess the impact of the guideline update of 31 August 2019, the cohort was divided in two groups. Group A (pre-guideline update) comprising of patients discharged from CR until 31 August 2019 and group B (post-guideline update) discharged from CR after 31 August 2019.

Analyses were conducted with SPSS version 22 (IBM, Armonk, NY, USA) and R Version 1.2.13 (RStudio, Boston, MA, USA). Patients who declined to sign written informed consent were excluded in accordance with the protocol approved by the local ethics committee (Swissethics, EKNZ, Basel, Switzerland, 2019-02057).

## 3. Results

### 3.1. Baseline Characteristics

From January 2017 to April 2021 a total of 1320 patients were screened. Patients without CAD (*n* = 354) and those who did not complete the rehabilitation program (*n* = 32) as well as patients with missing LDL-C (*n* = 59) were excluded, leaving 875 patients for analysis (Appendix A). Group A (pre-guideline update) consisted of 593 patients (median age 62.3 years, 16% female) and group B (post-guideline update) consisted of 282 patients (median age 63.2 years, 24% female). Compared to group A, patients in group B were more often female (24% vs. 16%, *p* = 0.009), had more frequent history of smoking (current or past smoker) (70% vs. 62%, *p* = 0.019) and were more often diagnosed with hypertension (70% vs. 62%, *p* = 0.036) and dyslipidaemia (72% vs. 56%, *p* < 0.001). Other cardiovascular risk factors, index diagnoses leading to CR or index therapy did not differ between both groups (Table 1). Prescription rates of combination therapy and maximum dose statins at CR entry and discharge for Group A and Group B.

### 3.2. Impact of 2019 ESC/EAS Dyslipidaemia Guidelines on Intensity of Lipid-Lowering Therapy and LDL-C Target Rate Achievement

Comparing group A and group B, the intensity of lipid lowering therapy was greater in group B, with more patients on a maximal statin dose at CR entry (16% vs. 11%, *p* = 0.044) and at CR discharge (20% vs. 9%, *p* < 0.001) and more patients receiving combination therapies at CR entry (18% vs. 6%, *p* < 0.001) and at CR discharge (51% vs. 17%, *p* < 0.001), Figure 1. Additionally, prescription rates for ≥50% MDD high-potent statins were higher in group B at CR discharge (87% vs. 80% *p* = 0.046), but not at CR entry (Appendix A). Statin associated symptoms leading to statin withdrawal or dose lowering were more frequent in group B compared to group A (6% vs. 2%, *p* < 0.001).

From CR entry to discharge a significant reduction of LDL-C (median 1.7, IQR 1.3–2.0 vs. median 1.4, IQR 1.1, 1.7, *p* < 0.001) could be observed in group B, whereas LDL-C in group A was similar at CR entry and at discharge (median 1.6, IQR 1.2–2.0 vs. median 1.5, IQR 1.2–1.9, *p* = 0.072) (Table 2). LDL-C distribution at entry in group B was similar to group A (median 1.7, IQR 1.3–2.0 vs. median 1.6, IQR 1.2–2.0, *p* = 0.31) but shifted to the lower end at discharge (median 1.4, IQR 1.1–1.7 vs. median 1.5, IQR 1.2–1.9, *p* < 0.001) (Table 2 and Appendix A).

#### LDL-C Changes at CR Entry and Discharge before (Group A) and after (Group B) 2019 ESC/EAS Lipid Guidelines

In group B the guideline targets of LDL-C were reached more often than in group A, this holds true for 2016 Guidelines (80% vs. 70%, *p* = 0.003) as well as 2019 Guidelines (53% vs. 39%, *p* < 0.001) (Table 2 and Figure 2).

### 3.3. Predictors of Therapy Intensification during CR

Since differences in therapy between group B and A were present at entry and at discharge, two logistic regression models to test for predictors leading to a more intensive lipid lowering therapy were conducted. In the first model the outcome was defined as an increase in statin dosage during CR revealing that dyslipidaemia (Coef 1.7, *p* < 0.001) and the patient being of group B (Coef 0.65, *p* = 0.027) had a significant positive relationship with an increase in statin dosage prescribed (Appendix A). Within the second model outcome was defined as newly begun therapy with ezetimibe during CR which revealed, similar to the first model, dyslipidaemia (Coef 0.66, *p* = 0.0018), smoking history (Coef 0.50, *p* = 0.019) and the patient being of group B (Coef 1.3, *p* < 0.001) were significant positively related to a newly begun therapy with ezetimibe (Appendix A). In contrast, no independent association with more intense therapy during CR was seen for other important patient characteristics such as age, sex, hypertension, as well as duration of CR or treatment modality (PCI versus CABG) in neither model (Appendix A).

### 3.4. Predictors of LDL-C Levels at the End of CR

In a multivariate quantile regression model age (Coef 0.0037, *p* = 0.048), a positive family history for cardiovascular events (ACS, stroke) (Coef 0.010, *p* = 0.021) as well as dyslipidaemia (Coef 0.13, *p* = 0.0012) and smoking history (Coef 0.14, *p* < 0.001) were independently associated with higher LDL-C levels. Notably, the patient being in Group B was the only significant predictor for lower LDL-C levels (Coef −0.21, *p* < 0.001) in this model. In contrast, no independent association with LDL-C levels was seen for important other patient characteristics such as sex, hypertension, medical therapy at entry, as well as duration of CR or treatment modality (PCI versus CABG) (Table 3).

## 4. Discussion

In this contemporary cohort of secondary prevention patients finishing a three month ambulatory CR program we found a significant impact of the 2019 ESC/EAS dyslipidaemia guidelines on prescription rates of combination therapies and statin doses, concomitant with an improved LDL-C target achievement rate. After the publication of the 2019 ESC/EAS dyslipidaemia guidelines prescription rates for ezetimibe tripled and the proportion of patients at highest recommended statin dose doubled. One in two patients was managed by combination therapy. This intensification of lipid-lowering therapy resulted in an achievement rate of 53% of patients reaching the LDL-C target of <1.4 mmol/L (<55 mg/dL) in the group finishing CR after the publication of the 2019 ESC/EAS dyslipidaemia guidelines with only 2% of patients having prescribed PCSK-9 inhibitors.

However, these changes are not only due to medication adaption during CR. Our results demonstrate that after publication of the 2019 guidelines patients at CR entry already are under more intense lipid-lowering therapy, which reflects the implementation of the guideline targets already in the acute hospital setting. Logistic regression reveals dyslipidaemia and a positive smoking history as independent predictors for increasing statin dosage or implementing ezetimibe therapy during the course of CR, indicating that individual risk profiles are taken into account when adjusting lipid-lowering therapies during CR.

To date, contemporary real world data regarding the impact of the 2019 ESC/EAS dyslipidaemia guidelines on lipid management is still scarce. Data from the first European multinational observational study since the 2019 guidelines, the SANTORINI study, reported a mean LDL-C level of 2.3 mmol/L in very high risk patients. In SANTORINI combination therapy was prescribed in 24.2% of patients (19.5% statin + ezetimibe; 4.7% statin + PCSK9i) [12]. In our contemporary cohort we report an overall combination therapy rate of 28% after CR, and 51% for those patients finishing the CR program after the publication of the 2019 ESC/ESA guidelines. Accordingly, the median LDL-C value of 1.5 mmol/L (1.4 mmol/L in patients after 2019 guidelines) reported in our cohort compares favorably to SANTORINI. Registries prior to 2019 such as EUROASPIRE V registry [7] and the PATIENT CARE registry [9] reported only 29% and 41.9% of patients reaching the then recommended <1.8 mmol/L (<70 mg/dL) LDL-C target. Even in the controlled setting of the ISCHEMIA trial, only 52% of patients reached the 2016 guideline targets [13].

The high LDL-C target achievement rate in our cohort may stem from a very high rate of high-intensity statin prescription in combination with ezetimibe. This treatment approach resulted in target achievement rates comparable to those reported by Koskinas et al. that modelled statin intensification effect and incremental ezetimibe effect on LDL-C levels in patients who were not on high-intensity statins or ezetimibe [14]. Furthermore, the structure of the 12-week CR program allows for close supervision with up-titration of lipid-lowering medication and permanent patient empowerment to increase statin therapy adherence as well as quick reaction to possible statin associated symptoms (SAS). Overall, we found a surprisingly low rate of permanent statin discontinuation after 3 months (3%), that is much lower than the reported 5–10% by literature [15,16]. However, SAS increased during the observation period after the 2019 guidelines due to a doubling of patients on maximal statin doses. The association of SAS with high statin doses is well known in clinical practice [17,18]. Thus, early combination of statins at moderate doses with ezetimibe and/or bempedoic acid should be preferred over maximizing statin monotherapy. In our cohort the overall use of PCSK9-inhibitors was very low (1.1%), which is explained by a national restriction in Switzerland for this therapy to patients with LDL-C >2.6 mmol/L despite optimal therapy. Given this restriction an additional of 33 patients (3.8%) would be eligible for PCSK9-inhibitor therapy at discharge from CR.

## 5. Limitations

There are several limitations to this study. First, LDL-C was calculated using the Friedewald formula. Although convenient, this calculation has several well-established limitations: (a) methodological errors may accumulate since the formula necessitates three separate analyses of TC, TGs, and HDL-C; and (b) a constant cholesterol/TG ratio in VLDL is assumed. With high TG values (>4.5 mmol/L or >400 mg/ dL) the formula cannot be used. This should especially be considered in non-fasting samples [6]. Second, this is a single center observational study performed in Switzerland and thus the results may not be representative for ambulatory CR programs in other health care systems. However, patient inclusion was consecutive and inclusive during a prespecified time period at a large ambulatory CR program covering the north-western part of Switzerland and thus best reflects current clinical practice in this setting. Third, the study provides no data on long term LDL-C target achievement rates and adherence to LDL-lowering therapies over time. The focus of this study was on the impact of the 2019 dyslipidaemia guidelines on LDL-C management in an ambulatory CR program of three months duration. Finally, due to a limited number of women included in the study, the results may not be representative for women. As in most other countries, low attendance rates for female patients in CR programs is also a clinical reality in Switzerland.

## 6. Conclusions

In conclusion we found a significant difference in prescription rates of lipid-lowering medication, especially combination therapies and statin doses, after publication of the 2019 ESC/EAS dyslipidaemia guidelines resulting in an LDL-C target achievement rate of >50% in CAD patients participating in ambulatory CR. Using high-intensity statins and increasing the use of combination therapies is crucial to further enhance the number of patients reaching the 2019 ESC/EAS LDL-C target goals.

## Figures and Tables

**Figure 1 jcm-11-07057-f001:**
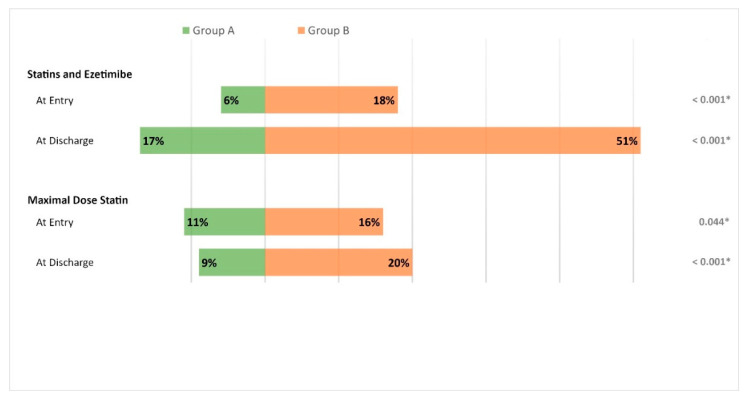
Mirror bar plot with prescription rates of combination therapy (statins and ezetimibe) as well as maximum dose statins at entry and discharge of cardiovascular rehabilitation (CR). The length of the columns represent the relative amount (in %) with regard to the entire cohort, green bars show **Group A**, orange bars represent **Group B**. Indication for significance = *p* Overall: 0.05 (indicated by *).

**Figure 2 jcm-11-07057-f002:**
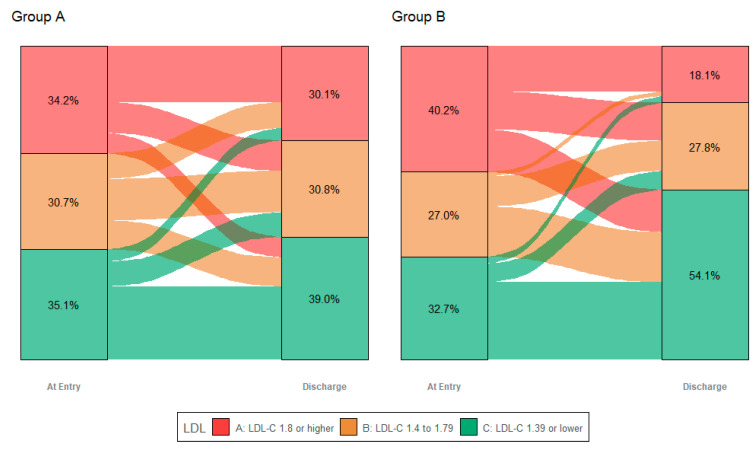
Alluvial plot with LDL-C at CR entry (left) and at discharge (right) for (**Group A**) and (**Group B**). The height of the columns represents the number of cases, the vertical bars visualize the target corridors of the ESC/EAS lipid guidelines 2016 and 2019 (green: <1.4 mmol/L, orange: <1–8 mmol/L). Y-Axis also shows the relative number of patients (in %) reaching the target corridors.

**Table 1 jcm-11-07057-t001:** Patient characteristics overall and stratified after discharge time period.

	Overall	Group A	Group B	*p*
	[875]	[593]	[282]	
Male Sex (%)	712 (81)	497 (84)	215 (76)	**0.009 ***
Age (median [IQR])	62.4 [54.9, 71.4]	62.3 [54.7, 70.8]	63.2 [55.4, 73.2]	0.23
BMI (median [IQR])	26.0 [24.0, 29.0]	26.0 [24.0, 29.0]	27.0 [24.0, 30.0]	0.074
Cardiovascular Risk Factors				
Diabetes Mellitus (%)	181 (21)	120 (20)	61 (22)	0.72
Hypertension (%)	566 (65)	369 (62)	197 (70)	**0.036 ***
Family History for CVE (%)	324 (37)	209 (35)	115 (41)	0.12
Dyslipidemia (%)	532 (61)	330 (56)	202 (72)	**<0.001 ***
Active Smoker (%)	138 (25)	94 (26)	44 (22)	0.42
Smoking History (%)	563 (64)	366 (62)	197 (70)	**0.019 ***
Diagnosis				0.73
STEMI	347 (40)	237 (40)	110 (39)	
NSTEMI	317 (36)	213 (36)	104 (37)	
Unstable Angina	24 (3)	14 (2)	10 (4)	
CCS	183 (21)	127 (21)	56 (20)	
Therapy				0.099
PCI	689 (79)	469 (79)	220 (78)	
CABG	114 (13)	83 (14)	31 (11)	
no therapy	71 (8)	41 (7)	30 (11)	

Categorical variables are described as *n* with % and continuous variables in median with interquartile range (IQR). Indication for significance = *p* Overall: 0.05 (indicated by bold letters and *); CABG: Coronary artery bypass surgery; CAD: Coronary artery disease; CCS: Chronic coronary syndrome; CVE: Cardiovascular Events; NSTEMI: Non-ST-elevation myocardial infarction; PCI: Percutaneous coronary intervention; STEMI: ST-elevation myocardial infarction.

**Table 2 jcm-11-07057-t002:** LDL-C levels at CR entry and at discharge overall and stratified after discharge time period.

	Overall	Group A	Group B	*p*
	[875]	[593]	[282]	
Patients with LDL-C at discharge <1.4 mmol/L (%)	351 (43)	232 (39)	119 (53)	**<0.001 ***
Patients with LDL-C at discharge <1.8 mmol/L (%)	592 (73)	412 (70)	180 (80)	**0.003 ***
LDL-C CR entry mmol/L (median [IQR])	1.6 [1.2, 2.0]	1.6 [1.2, 2.0]	1.7 [1.3, 2.0]	0.311
LDL-C CR discharge mmol/L (median [IQR])	1.5 [1.2, 1.8]	1.5 [1.2, 1.9]	1.4 [1.1, 1.7]	**<0.001 ***
	**<0.001 ***	0.072	**<0.001 ***	

Continuous variables are described in median with interquartile range (IQR); Indication for significance = *p* Overall: 0.05 (indicated by bold letters and *); LDL-C: Low-density lipoprotein cholesterol.

**Table 3 jcm-11-07057-t003:** Quantile Regression Weights for LDL-C at CR discharge.

	Coefficient/Estimate	95% Confidence Interval	*p*
Intercept	**1.0**	[0.60, 1.4]	**<0.001 ***
Age	0.0037	[−0.00024, 0.0074]	**0.048 ***
Male Sex	−0.049	[−0.17, 0.075]	0.44
Hypertension	0.0037	[−0.078, 0.086]	0.93
Dyslipidaemia	0.13	[0.053, 0.21]	**0.0012 ***
Smoking History	0.14	[0.057, 0.22]	**<0.001 ***
Statin Monotherapy	−0.049	[−0.24, 0.14]	0.61
Combination-therapy	−0.13	[−0.37, 0.11]	0.28
Therapy PCI	−0.0038	[−0.19, 0.18]	0.97
Therapy CABG	0.15	[−0.059, 0.37]	0.16
Group B	−0.21	[−0.30, −0.12]	**<0.001 ***
Rehabilitation Duration	0.0015	[−0.00026, 0.0033]	0.075
Family History for CVE	0.010	[0.016, 0.18]	**0.021 ***

Multivariate quantile regression model identifying factors associated with LDL-C at CR discharge. Positive coefficients/estimates indicate a positive correlation between above mentioned independent factors and LDL-C at discharge. Indication for significance = *p* Overall: 0.05 (indicated by bold letters and *); CABG: Coronary artery bypass surgery; CVE: Cardiovascular Events; PCI: Percutaneous coronary intervention.

## Data Availability

Not applicable.

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
