# Peer review of "Achievement of Low-Density Lipoprotein Cholesterol Targets in Cardiac Rehabilitation: Impact of the 2019 ESC/EAS Dyslipidaemia Guidelines"

_jcm, 2022, doi:10.3390/jcm11237057_

Round 1

Reviewer 1 Report

This study analyzed registration data of the ambulatory cardiovascular rehabilitation program of a University Hospital. The results showed a significant difference in prescription rates of lipid-lowering medication, especially combination therapies and statin doses, after publication of the 2019 ESC/EAS dyslipidaemia guidelines with an improvement of over 50% LDL-C target achievement, which is <1.4 mmol/L for CAD patients. As prescribed in the introduction, European Society of Cardiology (ESC) and the European Atherosclerosis Society (EAS) had lowered the LDL-C target values from <1.8 mmol/L (<70 mg/dL) to <1.4 mmol/L (<55 mg/dL) for secondary prevention of cardiovascular disease (CVD) in their 2019 dyslipidaemia guidelines. This study provides evidence for a positive impact from issuing of the ESC/EAS guideline. 

The study design was sufficiently described and the discussion/conclusions were justified by the results. 

Only few comments as the following:

What are the reasons for different patient characteristics between Group A and Group B? 

Line 22, The sentence should not be initiated by digits. 

Typos: Line 209, stating -> starting? 

Author Response

Response to Reviewers’ comments on manuscript JCM-2021032

We thank the Editors and the Reviewers for their time and effort given to the consideration of our manuscript and for their important comments, which helped to further improve the quality of the manuscript.

Please find a point-to-point response addressing the remaining concerns of the Reviewers below. All changes made in the manuscript are highlighted using track-change mode, the specifications for lines and pages refer to the track-change mode.

Ad Reviewer #1:

We thank the Reviewer for her/his review and the comment regarding the 1) difference in patient characteristics between patient group A and group B as well as 2) writing style. In the revised manuscript we address this comment as follows:

  • In contrast to group A, in group B the prevalence of arterial hypertension, dyslipidemia and a smoking history is higher. This may reflect the constant increase of cardiovascular risk factors in patients with coronary artery disease in Switzerland over time, which has been already shown in the swiss-wide AMIS plus registry (Cimci et al, 2021, PMID: 33258133). In both groups female patients are underrepresented, although being more frequent in group B. This increase could depict a rising awareness of the female gender gap, when it comes to diagnosing and managing cardiovascular disease (Vogel et al., 2021, PMID: 34010613).
  • We also addressed your stylistic remarks regarding line 22 (Now beginning with “In total,”) and line 209 (replacing “stating” for “statin”).

Reviewer 2 Report

Review ID: jcm-2021032.

Achievement of low-density lipoprotein cholesterol targets in cardiac rehabilitation: Impact of the 2019 ESC/EAS dyslipidae- mia guidelines “.

Very interesting study, which shows that in Switzerland lipid lowering treatment of patients with coronary heart disease according to guidelines may be more established than in other countries.

Major

·  Unfortunately, definition of “maximal dose statin” is not included in the manuscript. According to the guidelines, patients with CHD are at very high risk and must be treated properly by high potent statins in optimal dosages which means at least Rosuvastatin 20mg or Atorvastatin 40mg. (Mach F, et al. 2019 ESC/EAS Guidelines for the management of dyslipidaemias: lipid modification to reduce cardiovascular risk. Eur Heart J. 2020 Jan 1;41(1):111-188. doi: 10.1093/eurheartj/ehz455. Erratum in: Eur Heart J. 2020 Nov 21;41(44):4255. PMID: 31504418.)

In the Supplement Table 3 and the Results section should be stated, how many patients received at least Rosuvastatin 20mg or Atorvastatin 40mg in comparison of entry and discharge in comparison of the groups (50 or 100% MDD is not a clear statement).

·   Table 3: Not clearly described in the Table legend, what this model means? Factors associated with LDL-C-levels below 1,4mmol/l at discharge?

If so, how to explain the association of smoking with a better LDL-C at discharge? Please discuss.

Minor

·       LDL-C was calculated using the Friedewald formula in the study. This is a limitation, since not standard in other developed countries:

“Although convenient, the Friedewald calculated value of LDL-C has several well-established limitations: (i) methodological errors may accumulate since the formula necessitates three separate analyses of TC, TGs, and HDL-C; and (ii) a constant cholesterol/TG ratio in VLDL is assumed.With high TG values (>4.5 mmol/L or >400 mg/ dL) the formula cannot be used. This should especially be considered in non-fasting samples.” (Mach F, et al. 2019 ESC/EAS Guidelines for the management of dyslipidaemias: lipid modification to reduce cardiovascular risk. Eur Heart J. 2020 Jan 1;41(1):111-188.)

 This should be included in the limitation section.

·  Is there a funding bias? Is Ezetemib after 2019 maybe no longer patent protected and available as a generic drug in Switzerland? If so, this fact  may also explain the substantial increase of Ezetemib treated patients in comparison of both groups. Please discuss.

Author Response

Response to Reviewers’ comments on manuscript JCM-2021032

We thank the Editors and the Reviewers for their time and effort given to the consideration of our manuscript and for their important comments, which helped to further improve the quality of the manuscript.

Please find a point-to-point response addressing the remaining concerns of the Reviewers below. All changes made in the manuscript are highlighted using track-change mode, the specifications for lines and pages refer to the track-change mode.

Ad Reviewer #2:

We thank the Reviewer for her/his thorough review and the important points she/he raised regarding 1) maximum dosage of lipid-lowering therapy, 2) adaption of results section and suppl. Table 3, 3) the regression analysis identifying factors associated with LDL-C at CR discharge, 4) the limitation of the Friedewald formula as well as an 5) potential funding bias.

In the revised manuscript we clarified and discussed these key issues as follows:

1) Inserted in line 82, page 2: Dosage of high-potent statins is stated as absolute number and relative percentage of maximum daily dosage (MDD); both 40 mg Rosuvastatin and 80 mg Atorvastatin were considered as 100% MDD.“

2) Inserted in line 155, page 5: “Also, prescription rates for ≥50% MDD high-potent statins were higher in group B at CR discharge (87% vs. 80% p=0,046), but not at CR entry (Suppl. Table 3 and 4).

Inserted in Table 3 and Suppl. Table 4: Number of patients receiving at least Rosuvastatin 20mg or Atorvastatin 40mg (≥50% MDD).

Inserted in table legend Suppl. Table 3 and 4: „40mg Rosuvastatin and 80 mg Atorvastatin is defined as 100% MDD.“

3) Inserted in line 209, page 7 (table legend table 3): Multivariate quantile regression model identifying factors associated with LDL-C at CR discharge. Positive coefficients/estimates indicate a positive correlation between above mentioned independent factors and LDL-C at discharge.”

As stated in chapter 3.4, page 6, line 200, a positive family history for cardiovascular events (ACS, stroke) (Coef 0.010, p=0.021) as well as dyslipidaemia (Coef 0.13, p=0.0012) and smoking history (Coef 0.14, p<0.001) were independently associated with higher LDL-C levels. Notably, the patient being in Group B was the only significant predictor for lower LDL-C levels (Coef -0.21, p<0.001) in this model. That those factors are taken into account clinically and are also associated with increased statin and ezetimibe prescription is shown in Suppl. Table 1 and Suppl. Table 2 and is discussed in line 228-232, page 7.

4) Inserted in line 265, page 8: “There are several limitations to this study. First, LDL-C was calculated using the Friedewald formula. Although convenient, this calculation has several-well established limitations: a) methodological errors may accumulate since the formula necessitates three separate analyses of TC, TGs, and HDL-C; and (b) a constant cholesterol/TG ratio in VLDL is assumed. With high TG values (>4.5 mmol/L or >400 mg/ dL) the formula cannot be used. This should be especially considered in non-fasting samples.“

5) Inserted in line 85, page 2: „For the analysis, all single- and double-agent drugs containing ezetemib, that are commonly available in Switzerland were considered.“

Reviewer 3 Report

This is a retrospective analysis of prospectively collected data from the Swiss Secondary Prevention Registry in patients with coronary artery disease, who completed the ambulatory cardiovascular rehabilitation program of the University Hospital Basel, Switzer land from January 2017 to April 2021. Authors evaluated the impact of the guideline publication, the cohort of 875 patients was split into a pre-Guideline 2019 group (A) and a post-Guideline 2019 group (B). At discharge, more patients in group B were on maximal statin doses (20% vs. 9%, p<0.0001) and on combination therapy with ezetimibe 23 (51% vs. 17%, p<0.0001) than in group A, which resulted in 53% of patients reaching the LDL-C target of <1.4 mmol/L in group B. Regression analysis revealed that dyslipidaemia and smoking history represent independent predictors for intensified lipid-lowering medication. In general, the paper is well written.

Author Response

Response to Reviewers’ comments on manuscript JCM-2021032

We thank the Editors and the Reviewers for their time and effort given to the consideration of our manuscript and for their important comments, which helped to further improve the quality of the manuscript.